# Transgender fathering: Children's psychological and family outcomes

**Agnès Condat**[1,2]*, **Grégor Mamou**[3], **Chrystelle Lagrange**[1], **Nicolas Mendes**[1,4],
**Joy Wielart**[1], **Fanny Poirier**[1], **François Medjkane**[5], **Julie Brunelle**[1],
**Véronique Drouineaud**[4], **Ouriel Rosenblum**[1,4], **Nouria Gründler**[1,4], **François Ansermet**[6],
**Jean-Philippe Wolf**[4,7], **Bruno Falissard**[8], **David Cohen**[1,9]

**1** Service de Psychiatrie de l'Enfant et de l'Adolescent, Hôpital Pitié-Salpêtrière, AP-HP, Paris, France,
**2** CESP INSERM 1018, ED3C, Université Paris Descartes, Paris, France, **3** Clinique Dupré, Fondation
Santé des Etudiants de France, Sceaux, France, **4** Service Biologie de la Reproduction–CECOS, Hôpital
Cochin, AP-HP, Paris, France, **5** Service de Psychiatrie de l'Enfant et de l'Adolescent, CHU de Lille, Lille,
France, **6** Service de Psychiatrie de l'enfant et de l'adolescent, Département de l'enfant et de l'adolescent,
Hôpitaux Universitaires de Genève, Geneva, Switzerland, **7** Université Paris Descartes, Paris, France,
**8** Inserm, U669, Paris, France, **9** Institut des Systèmes Intelligents et de Robotiques, Université Pierre et
Marie Curie, Paris, France

* agnes.condat@aphp.fr

pone.0241214

Occidentale, FRANCE

**Data Availability Statement:** Data cannot be
shared publicly because of ethical concerns related
to the anonymity of participants. Data are available
from the Ethics Committee CERES (ethique.

## Abstract

Medical advances in assisted reproductive technology have created new ways for transgender persons to become parents outside the context of adoption. The limited empirical data does not support the idea that trans-parenthood negatively impacts children's development. However, the question has led to lively societal debates making the need for evidence-based studies urgent. We aimed to compare cognitive development, mental health, gender identity, quality of life and family dynamics using standardized instruments and experimental protocols in 32 children who were conceived by donor sperm insemination (DSI) in French couples with a cisgender woman and a transgender man, the transition occurring before conception. We constituted two control groups matched for age, gender and family status. We found no significant difference between groups regarding cognitive development, mental health, and gender identity, meaning that neither the transgender fatherhood nor the use of DSI had any impact on these characteristics. The results of the descriptive analysis showed positive psycho-emotional development. Additionally, when we asked raters to differentiate the family drawings of the group of children of trans-fathers from those who were naturally conceived, no rater was able to differentiate the groups above chance levels, meaning that what children expressed through family drawing did not indicate cues related to trans-fatherhood. However, when we assessed mothers and fathers with the Five-Minute Speech Sample, we found that the emotions expressed by transgender fathers were higher than those of cisgender fathers who conceived by sex or by DSI. We conclude that the first empirical data regarding child development in the context of trans-parenthood are reassuring. We believe that this research will also improve transgender couple care and that of their children in a society where access to care remains difficult in this population. However, further research is needed with adolescents and young adults.

clinique@aphp.fr) for researchers who meet the criteria for access to confidential data.

**Funding:** This project received funding from the Pfizer France Foundation and the Centre d'Activités et de Recherche en Psychiatrie Infanto-Juvénile (CARPIJ). The funders had no role in study design, data collection and analysis, decision to publish, or preparation of the manuscript.

**Competing interests:** The authors have read the journal's policy and have the following potential competing interests: AC has received funding for this project from the Pfizer France Foundation and the Centre d'Activités de Recherche en Psychiatrie Infanto-Juvénile (CARPIJ). Pfizer is a commercial funder. This does not alter our adherence to PLOS ONE policies on sharing data and materials. None of the authors have any commercial relationship with Pfizer, and there are no patents, products in development or marketed products associated with this research to declare.

## Introduction

For years, individuals and couples with fertility issues have been able to conceive thanks to biomedical technology advances [1]. These medical advances in assisted reproductive technology (ART) have created new ways for transgender persons to become parents outside the context of adoption [2]. Becoming a parent is a major life experience for human beings, and many trans people want to become parents [2]. However, hormone/surgical treatments that can benefit transgender people are potentially sterilizing. Currently, it remains technically impossible to develop the capacity to procreate after gender affirming surgery that involves removal of gonads, but it is possible to help trans people use their own gametes with assisted reproductive technology (ART). Trans parenthood is therefore possible through adoption but also through other paths, including using trans person's own gametes. Therefore, the World Professional Association for Transgender Health (WPATH) recommends discussing fertility options with patients before any treatment or medical and/or surgical interventions [3].

Heterosexual couples in which the male partner is a transgender man can turn to ART through artificial insemination with donor sperm insemination (DSI). For these couples, other possibilities are theoretically available. If the transgender man has self-conserved his oocytes/ovarian tissue before transition, or has retained his uterus and ovaries, the couple may consider using his eggs or ovarian tissue for his uterus-bearing partner, or he may consider pregnancy. Transgender women who have a male partner may ask for the help of a surrogate mother in countries where this is allowed and if this surrogate is not also a donor of her own oocytes for egg donation. In the first case, if the transgender man has self-conserved his oocytes/ovarian tissue before the transition, the couple may consider crossover in vitro fertilization (IVF) meaning that the transgender man provides oocytes that are microinjected with sperm of a male donor to obtain embryos that are transferred to his partner. In the second case, the preserved sperm of the transgender woman can similarly be used to fertilize donated oocytes for all or part of the offspring. Other combinations are possible in the case of gay couples [4]. However, these options are not available in many countries, as they are framed by laws and bioethics regulations, which can vary from one country to another.

In France, the transgender community is facing ostracism and each progress comes after long and conflicting debates in the society. For example, the law prohibits surrogacy, and ART was granted to heterosexual couples only until the House of Representatives changed the law in October 2019. The Senate still has to approve the new law, and ART will soon be open to couples of women and to single women, but this recent revision does not take into account the situation of transgender persons. Currently, the only option available to trans people in France is DSI for heterosexual couples in which one of the partners is a transgender man. The number of transgender people and transparent families has been estimated in France from the number of transitional surgeries counted by the national health insurance fund, leading to a very low and unreliable prevalence between 1/10,000 and 1/50,000 as not all trans people will undergo surgical transition [5]. However, the desire of transgender people to become parents is significant in France: 10% for transgender men and 4% for transgender women, in addition to the 20% of transgender people who want to adopt a child [6]. It should be noted, however, that in this study, 8% of transgender men and 45% of transgender women were already parents before their gender transition [7].

For many traditional individuals in France and in some ranges of the political and cultural society, biotechnological advances have also introduced new changes in the ancestral logic of conceiving that refer to two equal lineages (that of the mother and that of the father) and the traditional transition steps from conception to adult sexuality with several cultural/symbolic invariants, including (1) the temporal order, (2) the certainty of one's mother, and (3) the

parallel link between sex and gender [4,8]. The traditional boundaries among gender identity, sexuality, conception, gestation, procreation, and filiation are deeply challenged. If the spread of contraception over the last fifty years has caused an effective separation between sexuality and procreation, the current disruptions in conservative thinking are going much further [4]. These advances are confusing to many and sometimes may create discomfort and apprehension. They have led to lively societal debates opposing two different views: bio-catastrophists, on the one hand, and techno-prophets, on the other hand [9]. The former think that science serves as a driving force bringing about apocalyptic times. It contributes to the destruction of norms and traditional modes and understandings of the meaning of life, with severe consequences for society and, ultimately, the resulting end of the human species. In contrast, the latter believe that science offers the promise of a paradisiac future, a new redemptive era with a pure incorporeal spirit emerging from thinking machines.

These new ways to conceive a child crystallize several ethical questions discussed by the ESHRE Task Force that proposed guidelines [10]. Finally, the ultimate challenge seems to be the well-being of the child to come. Some may question the welfare of the offspring because some may think such trans-parenthood may be confusing for children [4].

The literature on the psychological well-being of transgender parents' children is sparse and limited to four studies that included 146 children [11–14]. Overall, they do not support the idea that trans-parenthood negatively impacts children's development. However, assessments were not blind, and children were born before their parent's transition. None of the children developed any gender identity variants, but some experienced difficulties in their relationship with peers (e.g., 33% in [13]); some suffered from depression (e.g., 13% in [14]). Notably, none of the previous studies have included a non-clinical control group or created a an experimental design specific to the research question.

An ideal situation to study the eventual impact of transgender parenthood is to explore children who are conceived by transgender people after their transition. In that case, children do not have to adapt to a new parental identity and are less confronted with socially aversive reactions [10]. Fifty-two children born between 2000 and 2015, by donor sperm insemination (DSI) in couples with a transgender man (assigned female at birth) and a cisgender woman, were followed qualitatively every two years. Children developed well without any major psychological morbidity [15]. Most of the child participants knew that they were born by third-party ART and that their fathers were assigned females at birth.

In the continuity of this last qualitative study, we proposed to carry out a two-year cross-sectional comparative study using standardized quantitative instruments. We aimed to compare cognitive development, mental health, gender identity, quality of life and family dynamics in children from transgender fathers conceived by donor sperm insemination (Trans-DSI group) and in two control groups matched for age, gender and family status (in couple vs separated): naturally conceived (NC group) children conceived by sex from cisgender parents and children from cisgender parents conceived by conventional DSI (Cis-DSI group). We chose to recruit two control groups to differentiate the impact of the transgender fatherhood from the possible impact of the use of ART in the condition of third-party procreation. In addition, there is also very few data on the overall development of children conceived with sperm donation and the current study is the first one in France.

A common intuitive idea in the general population and sometimes argued by many childhood professionals in France is that parents' perceptions and emotional experiences of their child vary according to their biological sex and their assigned sex during their own development. Many argue that having a transgender parent could have consequences in terms of parent-child relationship and family dynamics [16]. To address this issue, we used the Five-Minute Speech Sample (FMSS) [17,18], which is an instrument assessing the emotional climate

within the family through attitudes and feelings expressed by a parent towards his/her child and termed expressed emotions. In addition, we developed a quasi-experimental design based on children's family drawings to investigate whether cues not captured by the questionnaire could be seen by blind raters [17]. With this quasi-experimental design, we intended to explore more subtle differences between children born from Trans-DSI (meaning a transgender father) and NC children born from natural conception from cis-gender parents who had sexual intercourse. We used a method that was previously developed to explore how traumatic experience could be guessed without explicit information through participants' responses from an experimental task using a permutation test [19]. Here, the task we proposed to children was drawing a family. We hypothesized that children born from Trans-DSI would not be more likely to use atypical representations during family drawing (e.g., of men/fathers and of sexual indices) to be detectable by external raters.

Based on the findings from the qualitative study, we hypothesized that (i) the psycho-affective development of children born by DSI whose father is a transgender man will not significantly differ from that of children born by conventional DSI or from that of children conceived by sexual intercourse of both cisgender parents (NC group); (ii) father and mother reports regarding their child will not differ; (iii) family dynamics in the families of children from Trans-DSI group will not significantly differ from that of families of children from Cis-DSI or from NC groups; using either a measure of parental expressed emotion towards the child; (iv) or through family drawing (meaning that no group of raters will be able to differentiate the family drawings between children from transgender fathers conceived by DSI and NC children).

## Methods and analysis

### Design

The study design is a monocentric cross-sectional comparative study over two years. The recruitment centre, CECOS-Cochin, is unique, but this centre is authorized to treat couples from all over France. When the first DSI for couples whose father was a transgender occurred in 2000, CECOS-Cochin was the only centre authorized in France. The study includes two domains: (i) a clinical assessment to assess psycho-affective development and (ii) a family exploration using parental expressed emotion towards the child and an experimental procedure (see below).

### Ethics

The protocol was approved by the CERES (*Comité d'Ethique de Recherche en Santé*) of Paris 5 University (Registration number: 2015/31). Informed written consent was obtained from parents and children for participation in the study, orally up to 11 years old and with a written document for parents and for children and adolescents aged 12 to 15 years old. Appropriate information was provided to parents and the child or adolescent according to their age. When parents did not give their child born by DSI information about how he/she was conceived, we respected their wish and did not reveal it. It was an important ethical concern that was addressed with the ethics committee. We ensured that the study was presented as a research on the psycho-emotional development of children according to their mode of conception where parents could tell their child they belong to the control group of children conceived "naturally" if it was their choice to do so. Children and adolescents were therefore informed of the general objectives of the study in which they would participate but not of their status among the three possible groups. The information was given to them as follows: *"Children are most often conceived by their parents, but parents may not be able to have children without the*

*help of doctors. This research tries to find out if the way a child has been conceived can have effects on the child's psychological and emotional development."*

## Recruitment of the participants

We constituted the first group (Trans-DSI group) composed of children of transgender fathers and cisgender mothers conceived by DSI based on the national cohort available [15]. Gender identity of the persons was established according to their statement during the interview conducted by the biologist and psychologists at CECOS-Cochin prior to conception by ART and was confirmed during our study's interview. All the mothers identified themselves as cisgender women and all the fathers as transgender men. All said themselves to be binary identified. We proposed the research to 30 families over the 37 families of the national cohort (53 children). Seven of the 37 families had children too young to participate in the study (less than 18 months, which is the threshold age for the instrument which assesses global psychopathology). Seventeen out of the 30 families (i.e. 57%) agreed to participate. Four families refused to participate saying they did not want to spend a day for research, and nine families could not be contacted due to invalid coordinates. In total, 32 children were included in our Trans-DSI group. One parent could provide ratings on several children (see description of the sample below). This number of children should theoretically make it possible to highlight a difference between a normal and pathological range on each of the subscales of the Child Behaviour Checklist, including the internalizing and the externalizing scores (see power calculation below).

Then, we recruited two control groups of the same size matched for age, sex and family status. These two control groups constitute convenience samples recruited according to the characteristics of the group of children born by DSI of transgender fathers. In the second group, children were born by conventional DSI of both cisgender parents (Cis-DSI group). Approximately 75 couples have a progressive pregnancy by DSI at CECOS-Cochin each year. It is common for these couples to come back with a second or third request for donor-insemination. During the years 2014 and 2015, it was proposed to couples consulting in this context and to all other couples with at least one child born by DSI who consulted the CECOS of Cochin for whatever reasons to participate in the research. We also contacted other couples who had given their written consent to be contacted again by the CECOS for a research programme at the time they received the DSI. We included 28 children in our Cis-DSI group. The third group included naturally conceived controls born by sexual intercourse of cisgender parents (NC group). They were recruited by announcement in the meetings of the departments concerned among the families of the professionals who agreed as well as in a public school in the neighbourhood.

The inclusion criteria encompassed a target population of children aged under 15 years who were born by DSI or conceived by sex for the NC group and who agreed to participate in a one-day evaluation with the consent of their parents. The exclusion criteria were (i) a poor understanding of written and/or spoken French that would not have allowed participants to correctly complete the questionnaires or pass the standardized interviews and (ii) a refusal from at least one parent to sign the consent form. All couples were heterosexual, as DSI is only available for heterosexual couples in France.

## Sample

Table 1 summarizes the sociodemographic and developmental characteristics of the participants. In total, we included 32 individuals in the Trans-DSI group, 28 individuals in the Cis-DSI group and 28 individuals in the NC group. The Trans-DSI group included 17 families: 7 had only one child included in the study, 6 had two children, 3 had 3 children and one family

**Table 1. Demographics and clinical characteristics of the participants.**

| | NC Group | Cis-DSI Group | Trans-DSI Group | Test | p (test) |
|---|---|---|---|---|---|
| | N = 28 | N = 28 | N = 32 | | |
| **Gender** | | | | | |
| Male | 21 (75%) | 16 (57%) | 25 (78%) | Chi2 | p = 0.096 |
| Female | 7 (25%) | 12 (43%) | 7 (22%) | | |
| **Socio-economics** | | | | | |
| Low | 1 (3%) | 0 (0%) | 1 (3%) | | |
| Intermediate | 2 (7%) | 20 (71%) | 16 (50%) | Fisher | **p<0.001** |
| Good | 25 (89%) | 8 (29%) | 15 (47%) | | |
| **Parental status** | | | | | |
| In couple | 28 (100%) | 24 (86%) | 29 (91%) | Fisher | **p = 0.038** |
| Separated | 0 (0%) | 4 (14%) | 3 (9%) | | |
| **Age in years** | | | | | |
| mean (SD) [min;max] | 7.63 (3.71) [1.3;14.2] | 4.31 (2.61) [0.7;11] | 6.25 (3.59) [1.6;13.9] | ANOVA | **p = 0.002** |
| **Total Intellectual Quotient ($N_{NC}$ = 26, $N_{CDSI}$ = 28, $N_{TDSI}$ = 32)** | | | | | |
| mean (SD) [min;max] | 118.7 (14.82) [97;153] | 111.2 (19.69) [81;151] | 113.9 (16.33) [93;150] | Kruskal-Wallis | p = 0.182 |
| **Gender Identity Interview for Children ($N_{NC}$ = 24, $N_{CDSI}$ = 19, $N_{TDSI}$ = 25)** | | | | | |
| **Total score** *[pr = 0;24]* mean (SD) [min;max] | 1 (2.12) [0;10] | 1.16 (1.86) [0;6] | 1.24 (1.45) [0;5] | Kruskal-Wallis | p = 0.45 |
| **Affective gender confusion** *[pr = 0;14]* mean (SD) [min;max] | 0.96 (1.99) [0;9] | 0.89 (1.45) [0;5] | 0.84 (1.14) [0;4] | Kruskal-Wallis | p = 0.87 |
| **Cognitive gender confusion** *[pr = 0;8]* mean (SD) [min;max] | 0.08 (0.28) [0;1] | 0.21 (0.63) [0;2] | 0.16 (0.8) [0;4] | Kruskal-Wallis | p = 0.72 |
| **Father CBCL T-Score ($N_{NC}$ = 27, $N_{CDSI}$ = 26, $N_{TDSI}$ = 30)** | | | | | |
| **Anxious/depressed** mean (SD) [min;max] | 52.19 (4.95) [50;70] | 51.31 (3.72) [50;68] | 53.33 (5.26) [50;66] | Kruskal-Wallis | p = 0.11 |
| **Somatic complaints** mean (SD) [min;max] | 52.56 (4.07) [50;62] | 51.42 (3.70) [50;62] | 53.8 (5.90) [50;72] | Kruskal-Wallis | **p = 0.043** |
| **Withdrawn/depressed** mean (SD) [min;max] | 52.81 (4.03) [50;66] | 51.65 (3.20) [50;63] | 54.4 (6.01) [50;70] | Kruskal-Wallis | p = 0.28 |
| **Attention problems** mean (SD) [min;max] | 52.04 (3.71) [50;66] | 52.35 (4.15) [50;64] | 52.43 (3.69) [50;62] | Kruskal-Wallis | p = 0.73 |
| **Agressive behavior** mean (SD) [min;max] | 51.85 (3.96) [50;63] | 52.12 (4.58) [50;69] | 52.5 (3.95) [50;63] | Kruskal-Wallis | p = 0.34 |
| **Internalizing** mean (SD) [min;max] | 45 (10.2) [29;65] | 41.65 (9.23) [29;63] | 47.87 (12.56) [29;69] | ANOVA | p = 0.11 |
| **Externalizing** mean (SD) [min;max] | 43.85 (9.82) [28;64] | 44.73 (10.88) [32;72] | 45.70 (10.05) [28;61] | ANOVA | p = 0.79 |
| **Total problems** mean (SD) [min;max] | 43.59 (10.31) [28;65] | 41.31 (9.98) [24;67] | 46.07 (11.29) [26;68] | ANOVA | p = 0.25 |
| **Mother CBCL ($N_{NC}$ = 27, $N_{CDSI}$ = 26, $N_{TDSI}$ = 31)** | | | | | |
| **Anxious/depressed** mean (SD) [min;max] | 52.89 (5.99) [50;74] | 52.23 (4.63) [50;68] | 52.06 (4.13) [50;64] | Kruskal-Wallis | p = 0.96 |
| **Somatic complaints** mean (SD) [min;max] | 52.3 (4.08) [50;64] | 51.77 (3.46) [50;64] | 53 (5.17) [50;70] | Kruskal-Wallis | p = 0.77 |
| **Withdrawn** mean (SD) [min;max] | 52.74 (4.46) [50;70] | 52.69 (3.94) [50;63] | 53.71 (4.70) [50;68] | Kruskal-Wallis | p = 0.45 |
| **Attention problems** mean (SD) [min;max] | 51.41 (3.04) [50;62] | 52.42 (4.69) [50;67] | 52.39 (3.77) [50;62] | Kruskal-Wallis | p = 0.68 |
| **Agressive behavior** mean (SD) [min;max] | 50.74 (2.01) [50;59] | 52.65 (4.92) [50;69] | 52.39 (3.49) [50;60] | Kruskal-Wallis | p = 0.06 |
| **Internalizing** mean (SD) [min;max] | 54.04 (10.43) [29;63] | 44.15 (10.01) [29;63] | 47.48 (9.47) [29;66] | ANOVA | p = 0.42 |
| **Externalizing** mean (SD) [min;max] | 43.41 (8.38) [28;58] | 47.31 (12.24) [28;72] | 45.68 (9.75) [28;61] | ANOVA | p = 0.38 |
| **Total_problems** mean (SD) [min;max] | 43.04 (8.14) [28;63] | 43.58 (11.23) [24;67] | 44.94 (9.5) [26;64] | ANOVA | p = 0.74 |

NC: Naturally Conceived; Cis-DSI: Conventional Donor Semen Insemination; Trans-DSI: Transgender father and Donor Semen Insemination; CBCL: Child Behavior Check List; GIIC: Gender Identity Interview for Children; N:Number of participants who completed the measure; pr: possible range.

had 4 children. The Cis-DSI group included 16 families of which 7 had only one child included in the study, 6 had two children and 3 families had 3 children. The NC group included 17 families, 7 of which had only one child included in the study, 9 had two children, and 1 family had 3 children. There was a large majority of boys (75%) in our study group due to the unexplained distribution in the Trans-DSI group.

In the Trans-DSI group, 29 out of 32 children (from 14 out of 17 families) knew that they had been conceived by DSI and 28 (from 15 out of 17 families) knew that their father had been

assigned female at birth. In total, only 3 transgender parents did not inform their children of their trans identity including 2 who did not inform the child about DSI. In the Cis-DSI group 25 out of 28 children (from 13 out of 16 families) knew that they had been conceived by DSI.

Although we tried to match all groups on parental status, age and gender, we did not succeed for age as children in the Cis-DSI group were significantly younger (t student: p<0.001 for Trans-DSI (mean age = 6.25) vs. Cis-DSI (mean age = 4.32); p = 0.019 for NC (mean age = 7.63) vs. Cis-DSI). However, there was no age difference between the Trans-DSI and NC groups. The younger age of the children in the Cis-DSI group was due to French DSI specificities. In contrast to many other European countries (e.g., Switzerland), sperm donation is anonymous and not telling children about DSI and sperm donation is a very common practice. It is only in more recent years that changes in practice have been noticed in parallel with ethical debates regarding the possibility for children to ask for genetic origin (meaning changing the status of anonymous donation) [20]. Consequently, we had difficulties finding parents who conceived with DSI and older children or adolescents accepting participation in the research. Additionally, socioeconomic status (SES) and family stability were higher in the NC group. However, it is notable that very few families (N = 2, 2.3%) across groups had low SES and that a small proportion of parents across groups (N = 7, 8%) were separated.

## Measures

Each child and family underwent a thorough single-day evaluation within the Department of Child and Adolescent Psychiatry at the Pitié-Salpêtrière Hospital in Paris or at home if parents wished to do so. The department is well suited for a one-day welcoming of children for these assessments. The evaluation day began with a family interview with a mental health professional using a semi-structured interview. The study explored several dimensions [17]. In order to test our first two hypotheses, we used respectively: (1) for cognitive development, an adapted rating according to age: for those 2 to 30 months of age, we used the Brunet-Lézine psychomotor development scale [21], which estimates a Developmental Quotient (DQ) based on the normative data available for 2-year-old French toddlers; for those 30 months to 6 years of age, the Wechsler Preschool and Primary Intelligence Scale for Children- third edition (WPPSI-3) [22], which is a standardized developmental test for preschool-age children that measures intelligence skills; and for those 7 years of age and older, we used the Wechsler Intelligence Scale for Children-fourth edition (WISC-4) [23], which is a standardized developmental test for school-age children that measures intelligence skills. (2) For mental health, we used Achenbach's Child Behaviour Checklist (CBCL) [24], which assesses global psychopathology for children aged 6 to 18 years. We used the Preschool version for children aged 1.5 to 5 years. Both versions have been validated in France, and our samples were compared with the French norms. The CBCL is a parent-report measure designed to record the behaviours of children (one questionnaire filled by the mother and one questionnaire filled by the father separately). Each item describes a specific behaviour, and the parent is asked to rate its frequency on a three-point Likert scale. The CBCL gives, among others, three main scores (Internalizing, Externalizing, and Total Problems): a T-score of 64 and above is considered clinically significant, values between 60 and 63 identify a borderline clinical range, and values under 60 are considered non-clinical. Only the scores that were assessed by both the 1.5–5 and the 6–18 scales were selected in our study and compared. (3) For gender identity, we used the Gender Identity Interview for Children (GIIC) [25]. The GIIC includes 12 questions scored using a 3-item Likert scale which assess affective and cognitive gender confusion within the child. The total score can range from 0 (normal score) to 24 (highest probability of GD) and when the score is above a threshold of 4 the child may be at risk of gender dysphoria. GIIC was the only

scale that was not validated in France. To ensure the properties of the French GIIC, we performed an interrater reliability study that found excellent Cronbach alpha (total score = 0.98, factor 1 = 0.95, factor 2 = 0.97) and a ROC analysis to discriminate children with gender incongruence from those who do not (on an independent sample of 25 children) that found the same threshold that the one recommended in the English version (threshold = 4 for possible ranges for GIIC total score within 0 and 24). (4) For quality of life, we used the Kidscreen 52 [26], which assesses the child's global quality of life. The Kidscreen 52 is an auto-questionnaire for children and young people from 8 years old and measures 10 health-related quality of life dimensions: Physical (5 items), Psychological Well-Being (6 items), Moods and Emotions (7 items), Self-Perception (5 items), Autonomy (5 items), Parent Relations and Home Life (6 items), Social Support and Peers (6 items), School Environment (6 items), Social Acceptance (Bullying) (3 items), and Financial Resources (3 items).

In order to test our third hypothesis, we used: (1) the Inventory of Parent and Peer Attachment (IPPA) [27], which is an auto-questionnaire for children and adolescents from 9 years old measuring various qualities of children's relationships with parents and peers, including trust, quality of communication, and feelings of anger and alienation. It contains three sub-questionnaires: one concerning the mother, one concerning the father and one concerning peers. (2) the Five-Minute Speech Sample (FMSS) [18], which assesses the emotional climate within the family through attitudes and feelings expressed by a relative of a family member termed expressed emotions. The FMSS is based on a material made by a relative towards the child during a recorded session with the following instruction: "*I would like to hear from you about your thoughts and feelings about (name of family member) in your own words and without any interruption on my part by questions or comments. When I ask you to start, I would like you to talk to me for 5 minutes, telling me what kind of person is (name of the family member) and how you get along with him/her. Once you start speaking, I prefer not to answer any questions before the end of the 5 minutes*". For each child we recorded separately one session with the mother and one session with the father. There is no specific age range for this instrument. The FMSS distinguishes two main categories of expressed emotions that measure levels of criticism and emotional over-involvement. Rating requires a specific training and each expressed emotion is rated low, intermediate or high. The critical dimension is based on the initial statement, the expressed relationship and the blame or the dissatisfaction supported, whereas the emotional over-involvement includes emotional displays, statements of attitude (e.g., extreme loving), self-sacrificing and overprotection or a lack objectivity, an excess of detail about the past, and more than five positive remarks regarding the child [28]. The number of occurrences of expressed emotions for each category is recorded and produces a percentage. The total score requires the use of algorithm combining the rating of the two categories. The interrater reliability for FMSS scoring was strong (kappa-critical EE = 0.64, p = 0.008), (kappa-emotional over-involvement EE = 0.68, p = 0.007), (kappa-EE total = 0.88, p<0.001).

## Guessing transgender fatherhood from children's family drawing

As said previously, to explore more subtle differences between children born from Trans-DSI (meaning a transgender father) and NC children, we used a method that was previously developed to explore how traumatic experience could be guessed without explicit information through participants' responses from an experimental task using a permutation test [19]. Here, the task we proposed to children was drawing a family. As permutation test offers binary combination of answers, the task was performed by only two groups: children from Trans-DSI and NC children. The task was inspired by the Corman's Family Drawing Test [29], which assesses the child's perception of family relationship. Drawing is a medium that offers to children the

possibility of working from the projective and symbolic value of their contents. Corman's Family Drawing Test examines the graphic level which considers the quality of the production (the line, its size, its strength, the pace of drawing and the space of the sheet used to make it), the formal level derived from the original studies of the "drawing of the good man" (the degree of development of the child through the representation of the body of the characters' drawn, the link between the different characters, as well as the different elements drawing), and the level of contents indicating a projective value of the drawing (unusual or anxious representations, specific psychological problems of each child, valorisation/devaluation of certain characters) [29].

Family Drawing promotes child's projection associated with what the family represents. We built our experiment on that ability. The experiment responds to an assumption commonly found in French society and among childcare professionals: having a transgender parent could influence the development of the child, his/her identity construction but also his/her representations especially with regard to the family [16]. In order to be able to capture these important points, we included a group of raters who were experienced child psychoanalysts expert in the interpretation of children's drawings (see below). The previous study using the same innovative method [19] showed that psychoanalyst evaluators were the only ones to recognize better and above chance adults who experienced a childhood trauma.

Raters with diverse experiences were asked to guess children's group by viewing the drawings. To explore which experiences in raters may be helpful, the family drawings were analysed by 20 raters (4 child and family psychoanalysts (FAMPSY), 4 adult psychiatrists (ADUPSY), 4 biologists working in ART (BIOL), 4 endocrinologists working with transgender individuals (ENDOC) and 4 students (STUD)). They were randomly shown the drawings and asked to blindly classify them according to whether the child had a transgender father using a 4-level Likert scale: *I am certain that the drawing was done by a child from the Trans-DSI group*, *I think that the drawing was probably done by a child from the Trans-DSI group*, *I think that the drawing was probably done by a child from the* NC *group*, and *I am certain that the drawing was done by a child from the* NC *group*. Differences between children's family drawings were evaluated with a generalization of the "lady tasting tea" procedure to link qualitative and quantitative approaches in psychiatric research [30].

## Power calculation

Given that we used two different methods, we had two power calculations to determine. For clinical assessment, as we hypothesized that the psycho-affective development of children born by DSI whose father is a transgender man will not significantly differ from that of children born by conventional DSI or from that of children conceived by sexual intercourse of both cisgender parents, we needed to ensure that the number of individuals included was high enough to ensure that if we had no differences between groups that the statistical power was sufficient. The minimum size of the sample was calculated to be able to show with an alpha error probability of 5% and a statistical power of 80% a significant difference between two groups on the CBCL, one of our main objectives. We used the Multicultural Supplement to the Manual for the ASEBA School-Age Forms & Profiles [31] baseline data that present mean scores and standard deviations for samples corresponding to the French population for each scale. We calculated for each scale the sample size needed to highlight a difference between the normal range and the clinical range defined for the scale with an alpha error probability of 5% and a statistical power of 80% given the reference mean scores and standard deviations for the French population. We used the statistical programme R, version 3.3.1 (R Foundation for Statistical Computing) [32] with the formula n.for.2means (m1, m2, sd1, sd2, ratio, alpha, power). The

minimum size was found between 3 and 24 per group according to the scale. Only the CBCL Total score required a minimum size of 38 per group.

For the experimental procedure exploring whether raters blind to children status could classify children's family drawings above chance levels, we used a permutation test based on a modified version of Fisher's *lady tasting tea* procedure [19,30,33]. This statistical procedure was chosen to limit type I error. The number of cases, controls and raters required to detect differences with a power greater than 80% for a p < .05 was calculated by Falissard et al. [30]. For a sensitivity and specificity of correctly categorizing each subject, both equal to 80%, 23 cases, 23 controls and 4 raters are enough to detect significant differences using the procedure described below with a type I error of .05 and a power calculated at 99% [30].

### Data processing and statistics

As requested by French regulation, all data were processed anonymously and confidentially. Data were identified only by a code number and correspondence between this code and the participant's name/surname could only be established through a private list kept separately in another office. We used the Pitié-Salpêtrière child psychiatry computerized database for the processing of these data (CNIL declaration n° 1303778). The data of the present study were analysed using the statistical programme R, version 3.3.1 (R Foundation for Statistical Computing) [32]. For each variable, statistics were summarized with numbers and percentages for qualitative variables and with means (standard deviations) or medians (quartiles) for quantitative variables.

The first analysis compared each variable across the three study groups: naturally conceived children (NC group), children conceived by conventional donor sperm insemination (Cis-DSI group) and children conceived by DSI from a transgender father (Trans-DSI group). Based on the qualitative exploratory study [15], we hypothesized no difference between the groups. For each quantitative variable, we explored data distribution and normality using visual exploration. When normality was not reached, we used the Kruskal-Wallis nonparametric test. When normality was reached, ANOVA was used for 3-group comparisons, followed by Student's t-test for 2-group comparisons. For qualitative variables, we used the chi-squared or Fisher exact test according to the number of values. No correction for multiple testing has been done since our main hypotheses were in favor of the null hypothesis.

The second analysis explored whether raters blind to children's status could classify children's family drawings above chance levels. This analysis was limited to two groups. We used a permutation test based on a modified version of Fisher's *lady tasting tea* procedure [28,30,33]. It is noticeable that since the raters know that half of the records belong to "cases" and the other half to "controls", the ratings cannot be considered as independent realizations of a random variable, such that a traditional Student t-test or Mann-Whitney test should not be used. In contrast, under the null hypothesis, cases' and controls' records are indistinguishable; all permutations of scores obtained for each record are equiprobable. Hence, a sound (one-sided) p-value can be estimated as the proportion of permutations of the n records for which the total score is higher or equal to the total score obtained in the experiment [30]. We used a two-sided p-value based on a similar principle here. Of note, because of multiple testing (5 totally separate p-values were computed empirically), the level for significance was p<0.01.

Therefore, the association between judges' ratings and the actual distribution of subjects into cases and controls was tested in the following way. First, a score was computed for each group of raters: FAMPSY, ADUPSY, BIOL, ENDOC and STUD. The score was obtained by summing all $4*46$ coded evaluations: +2 when the raters correctly answered yes or no, +1 when they correctly answered probably yes or probably no, –1 when they incorrectly answered

probably yes or probably no, and –2 when they incorrectly answered yes or no. Thus, for each rater, the score could vary from +92 for all correct guesses (with a maximal certainty) to –92 for no correct guesses. For each group of raters, the score could range from +368 for all perfects to –368 for maximum failure.

To determine whether a group classified cases and controls better than could have been expected by chance, a permutation test was performed as described above using R software version 3.3.1 (R Foundation for Statistical Computing) [32]. The p-value was finally equal to twice the number of permutations for which the score was above the score obtained for the original data set in the experiment. Given that we used a modified version of the lady tasting tea procedure, it is not possible to provide a table showing the number of permutations for each level of performance on the dyads because first, there are 4 judges and second, possible answers are not yes or no but +2, +1, -1 and -2. Therefore, the number of possible errors ranges between 0 and 92 (2*46).

## Results

### Developmental characteristics of the participants

As shown in Table 1, we found no significant differences between the 3 groups regarding general intelligence (118.7 (±14.82) vs. 111.2 (±19.69) vs. 113.9 (±16.33) for NC, Cis-DSI and Trans-DSI, respectively, Kruskall-Walis, p = 0.182), gender identity (1 (±2.12) vs. 1.16 (±1.86) vs. 1.24 (±1.45) for NC, Cis-DSI and Trans-DSI, respectively, Kruskall-Walis, p = 0.45), or overall mental status as assessed with the CBCL (e.g., mother's CBCL total problems: 43.04 (±8.14) vs. 43.58 (±11.23) vs. 44.94 (±9.5) for NC, Cis-DSI and Trans-DSI, respectively, ANOVA, p = 0.74). Only the Father's Somatic complaints T-Scores showed a significant difference between the three groups (p = 0.043). Two-to-two comparisons showed that the Trans-DSI group T-scores were significantly higher than those of the Cis-DSI group (Wilcoxon p = 0.013). There was no significant difference between the Trans-DSI and the NC group or between the Cis-DSI group and the NC group. However, the means of the T-scores in each of the 3 groups were neither in the pathological zone nor in the limit zone. Regarding CBCL scores on an individual level, one 11-year-old girl from the Cis-DSI group had an externalizing CBCL subscore reaching 70 the pathological threshold. Father's CBCL Total score as Mother's CBCL Total score was in the clinical range for one child in the Cis-DSI group. Father's CBCL Total score was in the clinical range for one child in the NC group and for three children in the Trans-DSI group, but Mother's CBCL Total score was in the normal range for all these children. In addition, no child or parent reported bullying or harassment during the semi-structured interview. The trans identity of the parent was known within his family and in most cases within his in-laws (15 out of 17 families). On the other hand, neither the friendship groups nor the professional or school environment for children was informed of trans identity.

As the groups were quite young, especially in the Cis-DSI group, we did not have enough data for the Kidscreen to perform statistical analysis. S1 Table summarizes the quality of life characteristics of the participants. Kidscreen was performed by 13 children and adolescents in the NC group, 10 in the Trans-DSI group and only 2 in the Cis-DSI group. Nevertheless, for the adolescents and older children able to complete this instrument, it appears that the preliminary data are reassuring as most indicated rather good quality of life.

We also wanted to investigate whether the father and mother in the different groups had a common view of their children's behaviour. The CBCL can be completed by each parent separately. We were able to obtain 29 pairs (father, mother) in the Trans-DSI group, 26 pairs in the Cis-DSI group and 27 pairs in the NC group. Then, we calculated the intra-class correlations

**Table 2. CBCL (Child Behavior Check List) and FMSS (Five Minutes Speech Sample): Correlation between father's and mother's scores.**

| | NC Group | Cis-DSI Group | Trans-DSI Group |
|---|---|---|---|
| | N = 28 | N = 28 | N = 32 |
| **CBCL T-score by Pairs (N_NC = 27, N_CDSI = 26, N_TDSI = 29)** | | | |
| **Anxious/depressed T-Score** ICC [lower bound;upper bound] | 0.39 [0.06;0.65] | 0.68 [0.44;0.83] | 0.63 [0.38;0.80] |
| **Somatic complaints T-Score** ICC [lower bound;upper bound] | 0.47 [0.15;0.70] | 0.80 [0.63;0.90] | 0.60 [0.32;0.78] |
| **Withdrawn depressed T-Score** ICC [lower bound;upper bound] | 0.79 [0.61;0.89] | 0.15 [-O.20;0.47] | 0.41 [0.08;0.66] |
| **Attention problems T-score** ICC [lower bound;upper bound] | 0.23 [-0.12;0.53] | 0.80 [0.62;0.89] | 0.01 [-0.33;0.36] |
| **Agressive behavior T-score** ICC [lower bound;upper bound] | 0.09 [-0.24;0.42] | 0.74 [0.54;0.87] | 0.20 [-0.15;0.51] |
| **Internalizing T-score** ICC [lower bound;upper bound] | 0.70 [0.48;0.84] | 0.66 [0.42;0.82] | 0.65 [0.40;0.81] |
| **Externalizing T-score** ICC [lower bound;upper bound] | 0.66 [0.41;0.82] | 0.81 [0.63;0.90] | 0.67 [0.43;0.82] |
| **Total problems T-score** ICC [lower bound;upper bound] | 0.66 [0.40;0.81] | 0.79 [0.60;0.89] | 0.64 [0.39;0.81] |
| **FMSS score by Pairs (N_NC = 19, N_CDSI = 25, N_TDSI = 30)** | | | |
| **Expressed Emotion** ICC [lower bound;upper bound] | 0.10 [-0.14;0.38] | 0.00 [-0.34;0.34] | 0.00 [-0.34;0.34] |
| **Criticism** ICC [lower bound;upper bound] | 0.20 [-0.15;0.51] | 0.16 [-0.19;0.48] | 0.67 [0.43;0.83] |
| **Emotional Over Involvement** ICC [lower bound;upper bound] | 0.18 [-0.09;0.46] | 0.02 [-0.33;0.36] | 0.00 [-0.34;0.34] |

NC: Naturally Conceived; Cis-DSI: Conventional Donor Semen Insemination; Trans-DSI: Transgender father and Donor Semen Insemination; ICC: Intraclass correlation; N: Number of participants who completed the measure.

between mother and father CBCL scores in each group (Table 2). We found that fathers globally responded as mothers for their child in the 3 groups looking at the internalizing, externalizing and total scores. Low correlation coefficients were only observed between fathers' and mothers' responses to certain domains (withdrawn/depressed in the Cis-DSI group and attention problems and aggressive/behaviour scales in the Trans-DSI group and in the NC group).

## Family dynamics

Inventory of Parents and Peers Attachment was performed by 11 children and adolescents in the NC group, 8 in the Trans-DSI group and only 2 in the Cis-DSI group. As the groups were quite young, especially in the Cis-DSI group, we did not have enough data to perform statistical analysis. S1 Table summarizes the attachment characteristics of the participants. It appears that these preliminary data indicate secure attachment.

Concerning parental expressed emotions toward the child, Table 3 summarizes the FMSS for mothers and fathers of each group. In contrast to children's characteristics, all parental expressed emotions showed significant differences across groups. Two-group comparisons (S2–S4 Tables) showed that Expressed Emotions, Criticism and Emotional Over-Involvement were significantly higher in Trans-DSI fathers than in both NC and Cis-DSI fathers. There was no significant difference between the Cis-DSI and NC groups in fathers' Expressed Emotion, Criticism and Emotional Over-Involvement. For mothers, we found that they responded differently across groups but given the distributions in percentages, which were neither homogeneous nor linear, we only conducted two-group comparisons: Expressed Emotions were higher in NC mothers than in Cis-DSI mothers, Criticism was higher in Trans-DSI mothers than in NC mothers, and Emotional Over-Involvement was higher in NC mothers than in both Cis-DSI and Trans-DSI mothers. Finally, we performed intra-class correlations between parent pairs (Table 2). We found that fathers and mothers responded differently in all three groups except for the critical dimension in the Trans-DSI group, where fathers' and mothers' responses were correlated (ICC = 0.67 [95%CI:0.43–0.83]).

**Table 3. Five Minute Speech Sample results.**

| | NC Group | Cis-DSI Group | Trans-DSI Group | Test | p (test) |
|---|---|---|---|---|---|
| | N = 28 | N = 28 | N = 32 | | |
| **Mother Expressed Emotion ($N_{NC}$ = 21, $N_{CDSI}$ = 28, $N_{TDSI}$ = 30)** | | | | | |
| Low | 1 (5%) | 9 (32%) | 3 (10%) | | **p = 0.005** |
| Limit | 13 (62%) | 13 (46%) | 15 (50%) | Chi2 | |
| High | 7 (33%) | 6 (21%) | 12 (40%) | | |
| **Mother Criticism ($N_{NC}$ = 21, $N_{CDSI}$ = 28, $N_{TDSI}$ = 30)** | | | | | |
| Low | 17 (81%) | 23 (82%) | 16 (53%) | | **p = 0.007** |
| Limit | 3(14%) | 4 (14%) | 9 (30%) | Chi2 | |
| High | 1 (5%) | 1 (4%) | 5 (17%) | | |
| **Mother Emotional Over Involvement ($N_{NC}$ = 21, $N_{CDSI}$ = 28, $N_{TDSI}$ = 30)** | | | | | |
| Low | 1 (5%) | 10 (36%) | 5 (17%) | | **p = 0.010** |
| Limit | 14 (67%) | 13 (46%) | 17 (57%) | Chi2 | |
| High | 6 (29%) | 5 (18%) | 8 (27%) | | |
| **Father Expressed Emotion ($N_{NC}$ = 19, $N_{CDSI}$ = 25, $N_{TDSI}$ = 31)** | | | | | |
| Low | 7 (37%) | 8 (32%) | 2 (6%) | | **p<0.001** |
| Limit | 10 (53%) | 15 (60%) | 17 (55%) | Chi2 | |
| High | 2 (10%) | 2 (8%) | 12 (39%) | | |
| **Father criticism ($N_{NC}$ = 19, $N_{CDSI}$ = 25, $N_{TDSI}$ = 31)** | | | | | |
| Low | 13 (68%) | 20 (80%) | 15 (48%) | | **p = 0.002** |
| Limit | 5 (26%) | 5 (20%) | 13 (42%) | Chi2 | |
| High | 1 (5%) | 0 (0%) | 3 (10%) | | |
| **Father Emotional Over Involvement ($N_{NC}$ = 19, $N_{CDSI}$ = 25, $N_{TDSI}$ = 31)** | | | | | |
| Low | 9 (47%) | 9 (36%) | 5 (16%) | | **p<0.001** |
| Limit | 9 (47%) | 14 (56%) | 17 (55%) | Chi2 | |
| High | 1 (5%) | 2 (8%) | 9 (29%) | | |

NC: Naturally Conceived; Cis-DSI: Conventional Donor Semen Insemination; Trans-DSI: Transgender father and Donor Semen Insemination; N:Number of participants who completed the measure.

## Are expert raters able to guess whether children have a transgender father when observing family drawings?

In the experimental procedure, we asked four groups of expert raters to guess whether children had a transgender father when studying family drawings. These experts were child and family psychoanalysts (FAMPSY), adult psychiatrists (ADUPSY), biologists working in ART (BIOL), and endocrinologists working with transgender individuals (ENDOC). To assess the possible framing effect [34], we also added a group of inexperienced raters (students STUD) who received simplified instructions. The results are summarized in Fig 1. No group of raters was able to distinguish, based on family drawings, children raised with a transgender father and conceived through DSI and donor sperm from NC children conceived by heterosexual parents and sexual intercourse. The details of each rater scoring and the calculation of guessing score by groups of raters are available in S5 Table.

## Discussion

To facilitate discussion of the current results, we propose to explore our 4 hypotheses. Our results validate our *first hypothesis* namely that psycho-affective development of children born by DSI whose father is a transgender man do not significantly differ from that of children born by

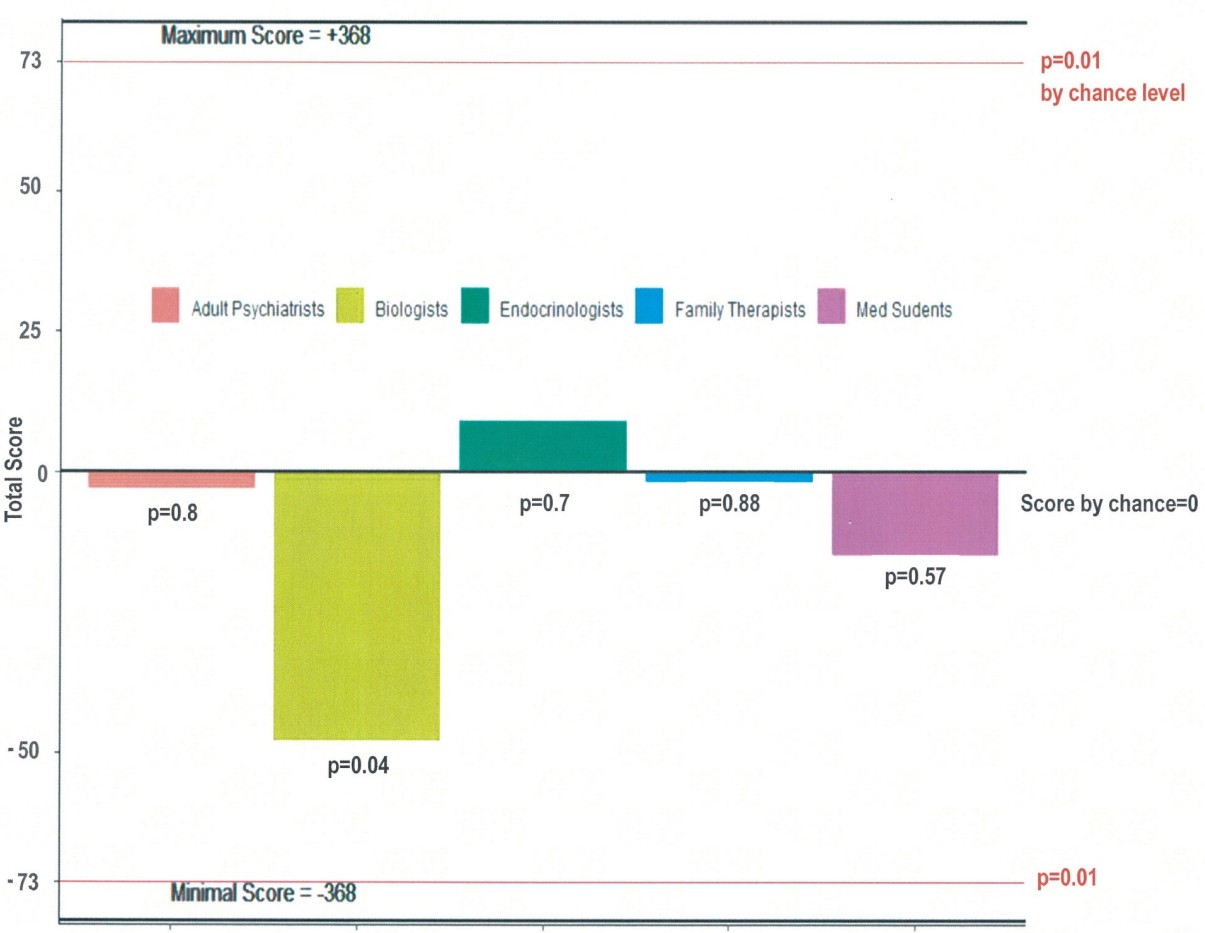

**Fig 1. The recognition scores of each rating group during the experimental procedure with the family drawings.** The scores calculated by adult psychiatrists, endocrinologists, biologists, family therapists and students when determining whether family drawings were done by children who had a transgender father and were born by DSI or by children who had a cisgender father and were born by sexual intercourse. For each group of raters, the score could vary from +368 for all perfect guesses to -368 for a complete failure, and the probability that the score differed from chance was calculated using a permutation test. The computed p-value for each group of raters is indicated on the bar (level of significance p = 0.01).

conventional DSI or from that of children conceived by sexual intercourse of both cisgender parents. We did not find any difference in cognitive development, gender identity, or mental health problems when comparing the three groups. Moreover, results regarding attachment and quality of life (descriptive analysis S1 Table) are overall reassuring. The cognitive development of all children showed no difference between groups, with an average IQ of 113.9 (ranges: 93–150).

In terms of psychopathology (as compared to the rates of psychopathology are among the (child) general population in France), three (10%) children whose father was transgender had a Father's CBCL Total score in the clinical Z-score range (total sore = 64; 65; 68) and one had a score in the limit zone. For all the children in the Trans-DSI group, including those children, Mother's CBCL Total scores were in the normal range. The current results contrast with those of White and Ettner's [14] study of children born before their parents' transition, as they found that 34% of the children had mental health disorders in a study population of 55 children, even if these numbers did not significantly exceed the rates found in the general population. In Freedman's study, 6% of the children were depressed or expressed suffering [13].

In previous studies investigating the psychopathology of children having a trans-parent, authors did not attribute disorders found to the fact that these children had a trans-parent. Our findings are consistent with this interpretation. It is likely that a mixture of risk factors intervened, such as bio-environmental factors, the child's own experience of his/her parent's transitioning or the social stigma of transitioning in terms of the children's environment (e.g., school and social media). Regarding bio-environmental factors, it should be noted that in White and Ettner's study, psychological disorders pre-existed the parent's transition in 63% of the included children [14].

In contrast with previous studies, our sample differs from previous studies. It was a non-clinical sample composed exclusively of children from transgender father who had their transition before the children were conceived by DSI. Information about DSI and/or father's transition was shared in most families. Fourteen out of 17 families had informed their children (28 out of 32 children) that their father was assigned female at birth and that they were born by DSI. Families were stable as only 3 out of 32 children (2 out of 17 families) had divorced parents. They were maintaining relationships with both parents. The samples from the previous studies are very different. They consisted mainly of children, one of whose parents was a transgender woman who had transitioned after the birth of the children [11–14]. They also included important recruitment bias: sample referred for expert opinion by court in Freedman's study [13] (in addition, this sample was not compared to a non-clinical sample of children but to a clinical sample of children and adolescents referred for gender issues); only parental interviews about their children in White and Ettner's study [14]; mainly children from conflicting/separated parents in Green, Freedman and White studies [9–14].

Quality of life was only assessed in a small number of children due to the young age in our cohort, but these descriptive results show no warning signs. In addition no child has been bullied or harassed in school or in social relationships. The risk of harassment and stigmatization for children of transgender parents is a potential risk [12,35]. Several studies have shown that peer harassment and teasing are infrequent i.e. Freedman et al [13] reported 33% of difficulties with peers in general, but no harassment or victimization; whereas Veldorale-Griffin [36] found 33% of bullying at school. Qualitative studies from parent-report data found that the children of transgender parents reported feeling protective towards their parent when they noticed discrimination or social rejection of their transgender parent [37–39]. The only qualitative study from an UK child-report data (33 children and adolescents of whom 23 were siblings) found that parental gender identity did not adversely impact upon how children and adolescents perceived their relationship with their parent as they expressed a sense of relational continuity. However, it might be otherwise relevant to their experiences both within and outside of the home with some of them expressing feelings of ambiguous loss, or the need to become responsible or to negotiate family display [40]. Also, many transgender parents were developing preventative strategies for themselves and their children not to be stigmatized [41]. Our results are in lines with previous studies exploring children of transgender parents. We believe that the low rate of harassment and stigmatization we found is related to the fact that children in our sample were born after the transition of their parent. It is likely that the absence of disclosure in the social environment outside the family circle contributed to prevent stigmatization and harassment.

We also note that all the children were cisgender identified. The interest in observing children's gender identity was to answer the question of whether having a transgender father has an impact on the development of the child's gender identity. One hypothesis could have been that having a transgender parent leads to a more fluid conception of gender with more possibilities for the child to identify outside the traditional heteronormative, cisnormative framework [42]. Our results do not support this hypothesis even if we must take into account the

small number of participants in view of the low prevalence of trans-identities in the general population (approximately 1% according to self-reported surveys [43,44]).

Regarding our *second hypothesis* (father and mother reports regarding their child do not differ) was addressed by comparing mothers and fathers CBCL scores of the same child. We found that fathers globally responded as mothers for their child in the 3 groups although differences were observed between fathers' and mothers' responses to certain domains (withdrawn/depressed in the Cis-DSI group and attention problems and aggressive/behaviour scales in the Trans-DSI group and in the NC group). It seems that neither gender identities, nor the transgender fatherhood nor the mode of conception, and therefore nor a genetic link between the father and the child, have an impact on the parent's view of the possible symptoms of their child.

Our *third hypothesis* was that family dynamics from children born by DSI whose father is a transgender man do not significantly differ from that of families with children born by conventional DSI or from that of families with children naturally conceived. Attachment could only be assessed in a small number of children due to the young age of our sample. But the descriptive results show no warning in this area (S1 Table). Regarding Expressed Emotions investigated with the FMSS, discussion should be cautious as expressed emotions have no specific psychopathological value *per se*. In the literature, FMSS Expressed Emotions or subscores were predictors of better outcome in young individuals with anorexia nervosa [27] or of poorer outcome in adult individuals with anorexia nervosa [45,46] or schizophrenia [47]. Key aspects for interpreting Expressed Emotions in families are the need (i) to distinguish fathers and mothers, (ii) to distinguish the type of condition as it is well known that chronic and severe conditions may impact family functioning [27,48–50], and (iii) to compare the same condition with the same instruments [51]. In our case, the current study is the first to explore parenting in the context of DSI. As transgender fathers differed from both the two other control groups, it seems that not the ART with donor insemination i.e. no genetic link between father and child, but the transgender fatherhood influenced the scores.

In contrast with our hypothesis, parental expressed emotion towards the child was not similar across fathers and across mothers. Indeed, we found that fathers and mothers responded differently in all three groups except for the critical dimension in the Trans-DSI group, where fathers' and mothers' responses were correlated. It seems, therefore, that the parent's gender identity has an impact on his/her expressed emotions towards the child.

We believe that the facts that (i) transgender fathers differed from both NC group fathers and Cis-DSI group fathers in terms of FMSS Expressed Emotions and (ii) transgender fathers were like their child's mothers in terms of criticism might have some meanings for trans-parenting. Indeed, by combining the objective difference of transgender fathers in the emotional experience with their child and the high intra-class correlation of critical dimension scores between transgender fathers and their wives (or paired mothers), several hypotheses can be formulated. (i) For each human being, becoming a parent is a self-flourishing experience transforming oneself and one's identity [51]. This experience is not the same for a transgender parent and triggers a different emotional mobilization after one's gender transition. (ii) Transgender parents must manage the projections of a heteronormative, cisnormative society on their way of making a family. They may cope to their unconventional situation with anxieties. These projections can also have an impact on the lived experience and emotional expression of these parents [4].

Our *last hypothesis* that was no group of raters would be able to differentiate the family drawings between children from transgender fathers conceived by DSI and NC children was validated by our results. Here, the quasi-experimental design based on children's family drawings investigated whether cues not captured by the questionnaire could be seen by blind raters [17]. Neither the child and family psychoanalysts' group nor the four other groups of raters

were able to differentiate the family drawings of children of trans-fathers from those of children in the NC group. For many clinicians, the "Drawing of the Family" is considered as a fine and sensitive means of evaluation of the different aspects of the child's functioning regarding his/her family. This experience also makes it possible to directly explore the children's family drawings while our psychopathological assessment was based on the CBCL where the questionnaires—given the young age of the children—are filled by the parents. The results show that there is no qualitative difference distinguishable by experts. Since one of our study limitations is the large age range (see below), we also performed a sensitivity analysis including only children aged 7 and older (2 groups of 11 children). At this age, all children can draw recognizable characters with gender-oriented cues [29]. The permutation test found the same results with a sufficient power [30]: no group of raters were able to guess whose children were sons or girls from transgender fathers (BIOL score = -38, p = 0.03; ENDOC score = 19, p = 0.36; STUD score = 0, p = 1; ADUPSY score = -6, p = 0.71; FAMPSY score = 8, p = 0.85; with guessing being linked to positive score only and p<0.01 being the level of significance after Bonferroni correction).

In France, few fertility service providers agree to perform DSI for couples whose man is transgender [15]. The main reason is the concern of professionals that echoes the concern of a part of society about the welfare of children to come [15]. At the same time, trans-people come to consultation before conceiving a child and ask practitioners about what science knows about possible risks for their children. We hope that the results of this study will enable professionals to demystify the issue of trans-parenting and that it will help reduce the anxiety of transgender people who are already parents or are parents in waiting.

Despite some strengths (including a unique sample of children with transgender fathers, two matched control groups, using standardized instruments, statistical power, and experimental procedures to investigate qualitative impressions in expressed emotion and family drawing), our study has several limitations. First, despite being the largest sample of children born from trans-fathers who had their transition before conception, the sample consisted of a small number of children. Similarly, we recruited a small number of families (only 17 out of 37 families solicited). Several other limitations result directly from the small size of this sample: (i) because of the small size of the Trans-DSI group, the two control groups constitute convenience samples recruited according to the characteristics of the Trans-DSI group. (ii) The age ranges and standard deviations are wide because it has not been possible to select a more homogeneous population of age that would have been at the same time sufficient for a statistical study. We cannot exclude the possibility of rare differences despite our power calculation. (iii) It should also be noted that we did not reach the theoretical size of the sample required for a power of 0.8 for the total score of the CBCL. This weakens our results although the theoretical number was reached for each sub score of the instrument. (iv) Our study group and consequently our matched control groups showed an over-proportion of boys (75%). This may be a viewed as a limitation. Nevertheless, despite matched comparisons between groups, the CBCL considers the gender of the child in the Z-score statistics that we used. Also, epidemiological studies show a 15% incidence of mental health in childhood and adolescence, slightly higher and earlier in boys [52]. This would rather tend to reinforce the validity of our results. Second, the study is transversal in nature and not prospective. Therefore, many children were young. This did not allow the evaluation of certain parameters, such as quality of life and self-reported attachment instruments. Third, despite our efforts to match participants for age, the NC and Trans-DSI groups were not perfectly matched for age to the Cis-DSI group. Fourth, the response rate was 56% in our study group (17 families out of 30). Further research is needed, especially with adolescents and young adults, as we cannot exclude that adolescence would eventually impact children's development.

**In conclusion,** we explored cognitive development, mental health, gender identity, and family dynamics in 32 children who were conceived by donor sperm insemination (DSI) in couples with a cisgender woman and a transgender man who had his transition before conception. We compared children's psychological and family outcomes in these children and in two matched control groups. Our study showed that the psycho-emotional development of children whose fathers are transgender is good and that there is no difference between these children and those of control groups. Similarly, no rater was able to differentiate the family drawings of children of trans-fathers from those of children in the NC control group. We also showed that the emotions expressed by transgender fathers who conceived by DSI were more intense than those of cisgender fathers who conceived by sex or by DSI. The generalization of our results should consider the limitations listed above but also the context of the sample that show middle/good SES and excellent family stability, two factors that contribute to children's mental health [53,54]. We believe that this research will also improve transgender couple care and that of their children in a society where access to care remains difficult in this population [55].

## Supporting information

**S1 Table. Kid-screen and attachment inventory of all participants who were eligible to perform the questionnaires.**
(PDF)

**S2 Table. Five Minute Speech Sample—NC group vs Trans-DSI group.**
(PDF)

**S3 Table. Five Minute Speech Sample—NC group vs Cis-DSI group.**
(PDF)

**S4 Table. Five Minute Speech Sample—Cis-DSI group vs Trans-DSI group.**
(PDF)

**S5 Table. Details of each rater scoring and calculation of guessing score by groups of raters who participated to the experimental procedure based on family drawing.**
(PDF)

## Acknowledgments

The authors thank all children and parents who participated in the study.

## Author Contributions

**Conceptualization:** Agnès Condat, Véronique Drouineaud, Jean-Philippe Wolf, Bruno Falissard, David Cohen.

**Data curation:** Agnès Condat.

**Formal analysis:** Agnès Condat, Grégor Mamou, Bruno Falissard, David Cohen.

**Funding acquisition:** Agnès Condat.

**Investigation:** Agnès Condat, Chrystelle Lagrange, Nicolas Mendes, Joy Wielart, Fanny Poirier, François Medjkane, Ouriel Rosenblum, Nouria Gründler, François Ansermet.

**Methodology:** Agnès Condat, Joy Wielart, Véronique Drouineaud, Jean-Philippe Wolf, Bruno Falissard, David Cohen.

**Project administration:** Agnès Condat.

**Software:** Agnès Condat, Grégor Mamou, Bruno Falissard.

**Supervision:** Bruno Falissard, David Cohen.

**Validation:** Agnès Condat, Grégor Mamou, Chrystelle Lagrange, Nicolas Mendes, Joy Wielart, Fanny Poirier, François Medjkane, Julie Brunelle, Véronique Drouineaud, François Ansermet, Jean-Philippe Wolf, Bruno Falissard, David Cohen.

**Writing – original draft:** Agnès Condat, Grégor Mamou, David Cohen.

**Writing – review & editing:** Agnès Condat, David Cohen.

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
