## [Decision Letter · Decision Letter 0]

21 Aug 2020

PONE-D-20-19647

Transgender fathering: children’s psychological and family outcomes

PLOS ONE

Dear Dr. Condat,

Thank you for submitting your manuscript to PLOS ONE. After careful consideration, we feel that it has merit but does not fully meet PLOS ONE’s publication criteria as it currently stands. Therefore, we invite you to submit a revised version of the manuscript that addresses the points raised during the review process including those which are raised by the second reviewer in the text itself (see attachments uploaded by reviewer 2

We look forward to receiving your revised manuscript.

Kind regards,

Michel Botbol, M.D.

Academic Editor

PLOS ONE

Journal Requirements:

'AC has received funding for this project from the Pfizer Foundation and the Centre d’Activités et de Recherche en Psychiatrie Infanto-Juvénile (CARPIJ). The sponsor did not intervene in the study design and did not participate in recruitment, analysis of the data or writing of the manuscript.'

We note that you received funding from a commercial source: Pfizer

5. Please include your tables as part of your main manuscript and remove the individual files. Please note that supplementary tables should remain as separate "supporting information" files.

6. Please include captions for your Supporting Information files at the end of your manuscript, and update any in-text citations to match accordingly. Please see our Supporting Information guidelines for more information: http://journals.plos.org/plosone/s/supporting-information

7. Your ethics statement must appear in the Methods section of your manuscript. If your ethics statement is written in any section besides the Methods, please move it to the Methods section and delete it from any other section. Please also ensure that your ethics statement is included in your manuscript, as the ethics section of your online submission will not be published alongside your manuscript.

Reviewers' comments:

Reviewer's Responses to Questions

**Comments to the Author**

1. Is the manuscript technically sound, and do the data support the conclusions?

Reviewer #1: Partly

Reviewer #2: Yes

2. Has the statistical analysis been performed appropriately and rigorously? 

Reviewer #1: Yes

Reviewer #2: I Don't Know

3. Have the authors made all data underlying the findings in their manuscript fully available?

Reviewer #1: Yes

Reviewer #2: Yes

4. Is the manuscript presented in an intelligible fashion and written in standard English?

Reviewer #1: Yes

Reviewer #2: Yes

5. Review Comments to the Author

Reviewer #1: Thank you for the opportunity to review this paper. The authors compare a sample of trans parent families (trans men and cis women with children conceived through donor insemination) with groups of donor insemination and unassisted conception families (both heterosexual-couple parent groups) on child adjustment and family process outcome measures. Very little empirical evidence exists on outcomes for children with trans parents. Trans parent families are growing in visibility and number and empirical data on trans parent families is much needed. In particular, studies using validated assessments of child adjustment (as used here) can make an important contribution, and I am not aware of any other studies with a French sample. I have several comments about the paper that may be of use to the authors.

General comments:

1. In order for the reader to be able to interpret the findings more accurately, more information is needed in the methods section about the measures used, e.g. what is the possible range of scores for each measure, and what was the scale reliability for each measure? How was the FMSS scored (is this a frequency count?) and what do higher scores indicate?

2. For clearer interpretation of the results it would be helpful to see all test statistics and effect sizes included throughout the results section (not just the p-values). There are several places where the authors say ‘very few’ or ‘most’ participants and it would be helpful here to have the Ns and %s included of how many children were in each category (lines 429, 441, 454).

3. I have concerns about the scientific rigour of the family drawing task. It’s not a measure of family environment that I’m particularly familiar with but as far as I’m aware it does not seem to have valid or reliable psychometric properties or to be associated with other child outcomes. I do understand that the measure may be read differently by different audiences and the authors suggest that it may be more recognised by clinicians, but my concern is that its inclusion may detract from the important findings elsewhere in the paper (i.e. those measured using more widely recognised and validated measures like the CBCL). The journal editors’ input would be useful here.

4. There are a several places where I would recommend slight changes to language to be more inclusive, for example the phrase ‘a transgender’ is used a couple of times. I would also suggest using ‘transgender fatherhood’ or ‘trans fatherhood’ rather than ‘the trans factor’. There are also some phrases which could be more idiomatic so I would suggest having the text read by a native English speaker before publication to aid clarity.

5. In a couple of places more context would be useful for readers who are not familiar with trans parent research or assisted reproduction research, e.g. the phrase ‘crossover IVF’ (line 110) may need to be explained. Also in line 121 it may be useful to explain why these estimates are unreliable/low (i.e. that not all trans people will undergo surgical transition).

6. I have an ethical concern about data availability and would recommend considering not making the data available. I am concerned that because the trans parent population is small, even if data are anonymised, participants may still be identifiable. For example, the dataset contains families with three and four children (which is fairly uncommon) – if a dataset also included information about family structure and demographic details the families may be identifiable if this information is put together.

Specific comments

1. line 33 ‘limited’ may be better than ‘paucity of’

2. Could the abstract include the information that this is a French sample?

3. In the introduction, could a little information be included to explain the social context for trans people in France (e.g. how much societal support/opposition is there?)

4. I’m not sure that paragraph three of the introduction (beginning line 126) is necessary. Keeping the article focused on children’s outcomes would make the introduction more streamlined.

5. In introducing hypothesis (i) and (ii) it would make sense to say ‘based on the findings from the first phase of the study’ to explain the hypotheses. The authors do explain this in other places in the article but including the information here would help with understanding these hypotheses. Could the rationale for hypotheses (iii) and (iv) be included?

6. Lines 423-424 – the findings from these three measures seem to be the most important in the paper. Can the finding for each measure be presented separately with the test statistic and p-value included in the text?

7. Line 435 – does ‘friendly environment’ mean friendship groups?

8. line 447 – what direction were the differences in?

9. lines 488 and 526 – findings are reported here that I think should also be in the results section

10. lines 519-521 – I think that these studies include parent-report data but not child-report data. Zadeh et al., 2019 might be a useful reference for child-report data.

11. When discussing the CBCL scores in the discussion, could the authors say what the rates of psychopathology are among the (child) general population in France so that the findings can be interpreted with some context?

12. When discussing the expressed emotion scores it would be useful to know why this is an important construct in family psychology (i.e. what outcomes is it associated with?). If trans parents have higher rates of expressed emotion, does this matter for children’s outcomes?

Reviewer #2: Review Comments to the Author

Please use the space provided to explain your answers to the questions above. You may also include additional comments for the author, including concerns about dual publication, research ethics, or publication ethics. (Please upload your review as an attachment if it exceeds 20,000 characters) (Limit 200 to 20000 Characters)

Thank you for this important contribution to transgender care and specifically reproductive care for transgender individuals

Please find attached a marked up copy of the manuscript

6. PLOS authors have the option to publish the peer review history of their article (what does this mean?). If published, this will include your full peer review and any attached files.

Reviewer #1: No

Reviewer #2: **Yes: **Jen Hastings, MD

---

## [Author Response · Author response to Decision Letter 0]

14 Sep 2020

RESPONSE TO EDITORIAL COMMENTS

1. Data available under request : Our choice was based on ethical restriction. Reviewer 1 think data should not be available at all since anonimity cannot be certain. See our response to his/her comment: Authors are very grateful with reviewer 1 for this comment that we definitively share. We felt that sharing data was compulsory in most journals and we did not check that Plos One offers the possibility not to share the data for individual protection. Therefore as suggested by reviewer 1 we followed this advice. It is now as follow in the revised version of the MS: ‘Availability of data and material: Because of ethical concerns, the data and material cannot be available. The trans parent population is small. Even if data are anonymized, participants may still be identifiable. For example, the dataset contains families with three and four children (which is fairly uncommon) and the families may be identifiable.’

Of course if the reviewer is wrong and the data needs to be available under request we can come back to our previous choice, the explaination is the one given in the revised MS. Contact for availability of the data will be Agnes Condat, MD, PhD agnes.condat@aphp.fr

2. Competing Interests Statement: AC has received funding for this project from the Pfizer Foundation and the Centre d’Activités de Recherche en Psychiatrie Infanto-Juvénile (CARPIJ). The sponsors did not intervene in the study design and did not participate in recruitment, analysis of the data or writing of the manuscript. 

Prizer is a commercial funder. None of the authors have any commercial relationship with Pfizer (employment, consultancy, patents, products in development, marketed products…). This does not alter our adherence to PLOS ONE policies on sharing data and materials.

3. Regarding the data not shown, the result of the permutation test is now given. Page 17 of the revised MS, it is now: “The permutation test found the same results with a sufficient power [30]: no group of raters were able to guess whose children were sons or girls from transgender fathers (BIOL score = -38, p=0.03; ENDOC score = 19, p=0.36; STUD score = 0, p=1; ADUPSY score = -6, p=0.71; FAMPSY score = 8, p=0.85; with guessing being linked to positive score only and p<0.01 being the level of significance after Bonferroni correction).”

4. Tables have been inculded in the main document

5. Captions for the supporting information are now given at the end of the document

6. The ethical statements was already in the method section.

RESONSE TO REVIEWERS

RESPONSE TO REVIEWER 1

Reviewer #1: Thank you for the opportunity to review this paper. The authors compare a sample of trans parent families (trans men and cis women with children conceived through donor insemination) with groups of donor insemination and unassisted conception families (both heterosexual-couple parent groups) on child adjustment and family process outcome measures. Very little empirical evidence exists on outcomes for children with trans parents. Trans parent families are growing in visibility and number and empirical data on trans parent families is much needed. In particular, studies using validated assessments of child adjustment (as used here) can make an important contribution, and I am not aware of any other studies with a French sample. I have several comments about the paper that may be of use to the authors.

We thank reviewer 1 for his/her encouragements and overall positive view on our MS. We particularly appreciated the feedbacks given. All issues were addressed in the revised version of the MS. We believe that thanks to both reviewers’ comments the MS is much improved.

General comments:

1. In order for the reader to be able to interpret the findings more accurately, more information is needed in the methods section about the measures used, e.g. what is the possible range of scores for each measure, and what was the scale reliability for each measure? How was the FMSS scored (is this a frequency count?) and what do higher scores indicate?

As requested we detailed the scoring method and the psychometrics of the two tests that are the least common: GIIC and FMSS. It is now page 9 of the revised MS: ‘For gender identity, we used the Gender Identity Interview for Children (GIIC) [25]. The GIIC includes 12 questions scored using a 3-item Likert scale which assess affective and cognitive gender confusion within the child. The total score can range from 0 (normal score) to 24 (highest probability of GD) and when the score is above a threshold of 4 the child may be at risk of gender dysphoria. GIIC was the only scale that was not validated in France. To ensure the properties of the French GIIC, we performed an interrater reliability study that found excellent Cronbach alpha (total score = 0.98, factor 1 = 0.95, factor 2 = 0.97) and a ROC analysis to discriminate children with gender incongruence from those who do not (on an independent sample of 25 children) that found the same threshold that the one recommended in the English version (threshold=4 for possible ranges for GIIC total score within 0 and 24).’

[…] and: ‘(2) the Five-Minute Speech Sample (FMSS) [18], which assesses the emotional climate within the family through attitudes and feelings expressed by a relative of a family member termed expressed emotions. The FMSS is based on a material made by a relative towards the child during a recorded session with the following instruction: “I would like to hear from you about your thoughts and feelings about (name of family member) in your own words and without any interruption on my part by questions or comments. When I ask you to start, I would like you to talk to me for 5 minutes, telling me what kind of person is (name of the family member) and how you get along with him/her. Once you start speaking, I prefer not to answer any questions before the end of the 5 minutes”. For each child we recorded separately one session with the mother and one session with the father. There is no specific age range for this instrument. The FMSS distinguishes two main categories of expressed emotions that measure levels of criticism and emotional over-involvement. Rating requires a specific training and each expressed emotion is rated low, intermediate or high. The critical dimension is based on the initial statement, the expressed relationship and the blame or the dissatisfaction supported, whereas the emotional over-involvement includes emotional displays, statements of attitude (e.g., extreme loving), self-sacrificing and overprotection or a lack objectivity, an excess of detail about the past, and more than five positive remarks regarding the child [28]. The number of occurrences of expressed emotions for each category is recorded and produces a percentage. The total score requires the use of algorithm combining the rating of the two categories. The interrater reliability for FMSS scoring was strong (kappa-critical EE=0.64, p=0.008), (kappa-emotional over-involvement EE=0.68, p=0.007), (kappa-EE total=0.88, p<0.001).’

2. For clearer interpretation of the results it would be helpful to see all test statistics and effect sizes included throughout the results section (not just the p-values). There are several places where the authors say ‘very few’ or ‘most’ participants and it would be helpful here to have the Ns and %s included of how many children were in each category (lines 429, 441, 454).

As requested, changes were made accordingly and the exact numbers are now reported. However, in a few occasions we did not repeat in the text all the data that appear in the tables as requested by PlosOne recommendations. 

3. I have concerns about the scientific rigour of the family drawing task. It’s not a measure of family environment that I’m particularly familiar with but as far as I’m aware it does not seem to have valid or reliable psychometric properties or to be associated with other child outcomes. I do understand that the measure may be read differently by different audiences and the authors suggest that it may be more recognised by clinicians, but my concern is that its inclusion may detract from the important findings elsewhere in the paper (i.e. those measured using more widely recognised and validated measures like the CBCL). The journal editors’ input would be useful here.

Authors understand the reviewer concern. The idea to use a stimulation (here the family drawing) and to have raters guessing based on the production of participants some characteristics of the participants was already used in other studies including one published in PlosOne (reference 19 in the revised MS). The second reviewer was very positive regarding this experiment. As reviewer 1 proposed, we believe that the Editor in charge will have an input.

4. There are a several places where I would recommend slight changes to language to be more inclusive, for example the phrase ‘a transgender’ is used a couple of times. I would also suggest using ‘transgender fatherhood’ or ‘trans fatherhood’ rather than ‘the trans factor’. There are also some phrases which could be more idiomatic so I would suggest having the text read by a native English speaker before publication to aid clarity.

We thank reviewer 1 for this comment. The text was edited for English by American Journal Expert (see certificate). However, we are aware that not being English fluent may be problematic. As requested we changed ‘trans factor’ for ‘transgender fatherhood’. We are also very grateful with reviewer 2 who kindly edited English in the MS. All changes were made accordingly.

5. In a couple of places more context would be useful for readers who are not familiar with trans parent research or assisted reproduction research, e.g. the phrase ‘crossover IVF’ (line 110) may need to be explained. Also in line 121 it may be useful to explain why these estimates are unreliable/low (i.e. that not all trans people will undergo surgical transition).

We thank reviewer 1 for this comment that was also shared by reviewer 2 who kindly suggested changes. It is now in the revised version of the MS page 4: ‘Currently, it remains technically impossible to develop the capacity to procreate after gender affirming surgery that involves removal of gonads, but it is possible to help trans people use their own gametes with assisted reproductive technology (ART).’ […] ‘In the first case, if the transgender man has self-conserved his oocytes/ovarian tissue before the transition, the couple may consider crossover in vitro fertilization (IVF) meaning that the transgender man provides oocytes that are microinjected with sperm of a male donor to obtain embryos that are transferred to his partner.’ […] ‘Heterosexual couples in which the male partner is a transgender man can turn to ART through artificial insemination with donor sperm insemination (DSI). For these couples, other possibilities are theoretically available. If the transgender man has self-conserved his oocytes/ovarian tissue before transition, or has retained his uterus and ovaries, the couple may consider using his eggs or ovarian tissue for his uterus-bearing partner, or he may consider pregnancy.’

Regarding the estimates it is now page 4 of the revised MS: ‘The number of transgender people and transparent families has been estimated in France from the number of transitional surgeries counted by the national health insurance fund, leading to a very low and unreliable prevalence between 1/10,000 and 1/50,000 as not all trans people will undergo surgical transition [5]’

6. I have an ethical concern about data availability and would recommend considering not making the data available. I am concerned that because the trans parent population is small, even if data are anonymised, participants may still be identifiable. For example, the dataset contains families with three and four children (which is fairly uncommon) – if a dataset also included information about family structure and demographic details the families may be identifiable if this information is put together.

Authors are very grateful with reviewer 1 for this comment that we definitively share. We felt that sharing data was compulsory in most journals and we did not check that Plos One offers the possibility not to share the data for individual protection. Therefore as suggested by reviewer 1 we followed this advice. It is now as follow in the revised version of the MS: ‘Availability of data and material: Because of ethical concerns, the data and material cannot be available. The trans parent population is small. Even if data are anonymized, participants may still be identifiable. For example, the dataset contains families with three and four children (which is fairly uncommon) and the families may be identifiable.’

Specific comments

1. line 33 ‘limited’ may be better than ‘paucity of’

The change was made accordingly.

2. Could the abstract include the information that this is a French sample?

The change was made accordingly.

3. In the introduction, could a little information be included to explain the social context for trans people in France (e.g. how much societal support/opposition is there?)

4. I’m not sure that paragraph three of the introduction (beginning line 126) is necessary. Keeping the article focused on children’s outcomes would make the introduction more streamlined.

We decided to join these two points in our response because they are related. We completely agree with reviewer 1 . The paragraph beginning line 126 was intended to explain the social context and its tensions in the case of France. We revised the MS to clarify this point.

5. In introducing hypothesis (i) and (ii) it would make sense to say ‘based on the findings from the first phase of the study’ to explain the hypotheses. The authors do explain this in other places in the article but including the information here would help with understanding these hypotheses. Could the rationale for hypotheses (iii) and (iv) be included?

We thank reviewer 1 for this comment that makes the introduction easier to follow. As requested we moved the rational that appeared in the methods to the introduction.

6. Lines 423-424 – the findings from these three measures seem to be the most important in the paper. Can the finding for each measure be presented separately with the test statistic and p-value included in the text?

The change was made accordingly.

7. Line 435 – does ‘friendly environment’ mean friendship groups?

We thank reviewer 1 for this comment. ‘Friendship groups’ was what we wanted to mean. The change was made accordingly.

8. line 447 – what direction were the differences in?

We thank reviewer 1 for this comment. Indeed the use of the word differences here was inappropriate as we are showing correlation in table 2. Sorry for this mistake. To clarify this issue in the revised version of the manuscript we changed the sentence as follow. It is now page 13 of the revised version of the MS: “Low correlation coefficients were only observed between fathers’ and mothers’ responses to certain domains”

9. lines 488 and 526 – findings are reported here that I think should also be in the results section

We thank reviewer 1 for this comment. IQ score were given in table 1. They are now also indicated in the text of the result section (see previous response minor point 6). Similarly, total score from the Gender Identity Interview is now also given in the text of the revised result section.

10. lines 519-521 – I think that these studies include parent-report data but not child-report data. Zadeh et al., 2019 might be a useful reference for child-report data.

We thank reviewer 1 for pointing Zadeh et al. study. This was very appropriate. We included a sentence to quote the interesting results of this study. It is now page 15 of the revised version of the MS: “Qualitative studies from parent-report data found that the children of transgender parents reported feeling protective towards their parent when they noticed discrimination or social rejection of their transgender parent [37-39]. The only qualitative study from an UK child-report data (33 children and adolescents of whom 23 were siblings) found that parental gender identity did not adversely impact upon how children and adolescents perceived their relationship with their parent as they expressed a sense of relational continuity. However, it might be otherwise relevant to their experiences both within and outside of the home with some of them expressing feelings of ambiguous loss, or the need to become responsible or to negotiate family display [40].”

11. When discussing the CBCL scores in the discussion, could the authors say what the rates of psychopathology are among the (child) general population in France so that the findings can be interpreted with some context?

The change was made accordingly (line 504 of the revised version of the MS).

12. When discussing the expressed emotion scores it would be useful to know why this is an important construct in family psychology (i.e. what outcomes is it associated with?). If trans parents have higher rates of expressed emotion, does this matter for children’s outcomes?

We thank reviewer 1 for this comment that should help clarifying how interpreting the concept of expressed emotions. We added the following sentences in the revised version of the MS. It is now page 16: “Regarding Expressed Emotions investigated with the FMSS, discussion should be cautious as expressed emotions have no specific psychopathological value per se. In the literature, FMSS Expressed Emotions or subscores were predictors of better outcome in young individuals with anorexia nervosa [27] or of poorer outcome in adult individuals with anorexia nervosa [44, 46] or schizophrenia [47].” The corresponding references were added.

 

RESPONSE TO REVIEWER 2

Reviewer #2: Review Comments to the Author

Thank you for this important contribution to transgender care and specifically reproductive care for transgender individuals

We thank reviewer 2 for his/her encouragements and overall positive view on our MS. We particularly appreciated the feedbacks given and obviously the kind commitment in editing English of our MS. Even if we paid a professional editing service recommended by most Publisher (including Plos), we are aware that given our topic some formulation may carry ambiguity or negativity from one language to another.

Please find attached a marked up copy of the manuscript

Again, thank you so much for being so precise on the reviewing. All changes except one were made accordingly to your proposals of rewriting or your queries when some ambiguous formulations needed clarifications. All changes are available in the version of the revised MS in which they appear in yellow. Only on one point we had to maintain our previous formulation, it is about the suggested change for the title, namely, to replace “trangender fathering” by “transgender parenting”. Despite the relevance of this suggestion on which we fully agree with you, we also had to consider the requests of reviewer 1 which were to be as precise as possible. Therefore we kept the title “transgender fathering” which more precisely indicates the subject of the study.

E.g. In the introduction section, we now give more information for readers that are not familiar with ART and transgender. It is now in the revised version of the MS: “Heterosexual couples in which the male partner is a transgender man can turn to ART through artificial insemination with donor sperm insemination (DSI). For these couples, other possibilities are theoretically available. If the transgender man has self-conserved his oocytes/ovarian tissue before transition, or has retained his uterus and ovaries, the couple may consider using his eggs or ovarian tissue for his uterus-bearing partner, or he may consider pregnancy. Transgender women who have a male partner may ask for the help of a surrogate mother in countries where this is allowed and if this surrogate is not also a donor of her own oocytes for egg donation. In the first case, if the transgender man has self-conserved his oocytes/ovarian tissue before the transition, the couple may consider crossover in vitro fertilization (IVF) meaning that the transgender man provides oocytes that are microinjected with sperm of a male donor to obtain embryos that are transferred to his partner.”

E.g. In the method section, we do not use any more experimental procedure as heading and in the MS. We decided to be ‘just descriptive’. It is now page 10 of the revised MS: “Guessing transgender fatherhood from children’s family drawing”

---

## [Editor Report · Decision Letter 1]

12 Oct 2020

Transgender fathering: children’s psychological and family outcomes

PONE-D-20-19647R1

Dear Dr Condat ,

We’re pleased to inform you that your manuscript has been judged scientifically suitable for publication and will be formally accepted for publication once it meets all outstanding technical requirements.

Kind regards,

Michel Botbol, M.D.

Academic Editor

PLOS ONE
---

## [Editor Report · Acceptance letter]

9 Nov 2020

PONE-D-20-19647R1 

Transgender fathering: children’s psychological and family outcomes 

Dear Dr. Condat:

I'm pleased to inform you that your manuscript has been deemed suitable for publication in PLOS ONE. Congratulations! Your manuscript is now with our production department. 

Kind regards, 

on behalf of

Pr Michel Botbol 

Academic Editor

PLOS ONE